# Yeast Nuak1 phosphorylates histone H3 threonine 11 in low glucose stress by the cooperation of AMPK and CK2 signaling

Seunghee Oh[1], Jaehyoun Lee[1], Selene K Swanson[1], Laurence Florens[1], Michael P Washburn[1,2], Jerry L Workman[1]*

[1]Stowers Institute for Medical Research, Kansas City, United States; [2]Department of Pathology and Laboratory Medicine, University of Kansas Medical Center, Kansas City, United States

**Abstract** Changes in available nutrients are inevitable events for most living organisms. Upon nutritional stress, several signaling pathways cooperate to change the transcription program through chromatin regulation to rewire cellular metabolism. In budding yeast, histone H3 threonine 11 phosphorylation (H3pT11) acts as a marker of low glucose stress and regulates the transcription of nutritional stress-responsive genes. Understanding how this histone modification 'senses' external glucose changes remains elusive. Here, we show that Tda1, the yeast ortholog of human Nuak1, is a direct kinase for H3pT11 upon low glucose stress. Yeast AMP-activated protein kinase (AMPK) directly phosphorylates Tda1 to govern Tda1 activity, while CK2 regulates Tda1 nuclear localization. Collectively, AMPK and CK2 signaling converge on histone kinase Tda1 to link external low glucose stress to chromatin regulation.

## Introduction

To ensure survival, cells must properly adapt to changes in available nutrients by altering their metabolism. This adaptation can be achieved through the cooperation of multiple metabolic pathways. Among those pathways, AMP-activated protein kinase (AMPK) signaling has a central role in energy homeostasis, especially when the cellular nutrient supply is low (*González et al., 2020*). AMPK orthologs are very well-conserved from yeast to human and occur as heterotrimeric complex comprising catalytic α subunit and regulatory β and γ subunits (*Ross et al., 2016*). For its proper function, the AMPK catalytic α subunit needs to be phosphorylated at its conserved threonine residue (threonine 172 in rat [*Hawley et al., 1996*] and threonine 210 in budding yeast [*McCartney and Schmidt, 2001*]) within the activation loop of its kinase domain. In budding yeast, the catalytic α subunit is sucrose non-fermenting 1 (Snf1). This name comes from its requirement for growth by sucrose fermentation (*Carlson et al., 1981*). The regulatory γ subunit of budding yeast AMPK is Snf4, and its interaction with Snf1 liberates Snf1 from auto-inhibition that interferes with Snf1 threonine 210 (T210) phosphorylation (*Chen et al., 2009*). Budding yeast encodes three AMPK β subunit proteins: Sip1, Sip2, and Gal83. They are partially redundant. Only when all three β subunits are deleted, cells exhibit growth defects in media containing ethanol or glycerol as a sole carbon source, as the *snf1Δ* mutant does (*Erickson and Johnston, 1993*; *Schmidt and McCartney, 2000*). The β subunits have a conserved C-terminal sequence that interacts with α and γ subunits (*Jiang and Carlson, 1997*; *Yang et al., 1994*) and a specific N-terminal sequence for each β subunit that confers a distinctive subcellular localization of the Snf1 complex. Notably, Gal83 is required for Snf1 nuclear localization upon glucose depletion (*Vincent et al., 2001*).

Snf1 interacts with and phosphorylates several proteins involved in nutritional stress responses (*Coccetti et al., 2018*). Its substrates include several transcription factors such as Cat8 and Sip4,

*For correspondence:
jlw@Stowers.org

which bind to carbon-source-responsive elements for the transcription of gluconeogenic genes (*Lesage et al., 1996*; *Randez-Gil et al., 1997*; *Vincent and Carlson, 1998*). Cat8 is required for proper expression of the transcription factor Adr1, which is also phosphorylated by Snf1 (*Kacherovsky et al., 2008*; *Young et al., 2002*). Snf1 also phosphorylates the Mig1 transcription repressor, which suppresses the transcription of glucose-repressive genes (*Nehlin and Ronne, 1990*). Mig1 phosphorylation by Snf1 activates its nuclear export signal leading to expulsion of Mig1 from the nucleus (*DeVit and Johnston, 1999*; *Papamichos-Chronakis et al., 2004*; *Treitel et al., 1998*). Interestingly, the yeast hexokinase 2 (Hxk2) protein interacts with Mig1 in the nucleus to protect Mig1 from phosphorylation by Snf1 (*Ahuatzi et al., 2007*). When glucose becomes scarce, Hxk2 is phosphorylated at serine 15, which inactivates its nuclear localization signal and, in turn, facilitates its cytoplasmic localization (*Fernández-García et al., 2012*; *Kriegel et al., 1994*). Snf1 had been believed to be a Hxk2 serine 15 kinase; however, recent studies revealed that yeast Nuak1 homolog Tda1 is responsible for Hxk2 phosphorylation (*Kaps et al., 2015*; *Kettner et al., 2012*). Thus far, Hxk2 is the only known substrate of Tda1, and how Tda1 activity is regulated upon low glucose stress is unknown.

AMPK also has a central role in regulating nutritional stress-specific chromatin modification (*Lee et al., 2020*). In yeast, Snf1 is required for H3 serine 10 phosphorylation (H3pS10) at the *INO1* gene promoter to regulate its transcription (*Lo et al., 2001*). Human AMPK phosphorylates H2B at serine 36 at the promoters of p53-responsive genes upon glucose starvation (*Bungard et al., 2010*). AMPK indirectly regulates H3 arginine 17 di-methylation by affecting histone arginine methyltransferase CARM1 protein level in the nucleus upon apoptosis-inducing conditions (*Shin et al., 2016*).

Recent studies imply that metabolic enzymes themselves also participate in chromatin regulation. Pyruvate kinase, a glycolysis enzyme, can phosphorylate H3 at threonine 11 in humans and in yeast (*Li et al., 2015*; *Yang et al., 2012*). Especially in yeast, pyruvate kinase forms a complex named SESAME with several metabolic enzymes involved in serine and SAM metabolism to regulate histone H3 threonine 11 phosphorylation (H3pT11) in glucose-rich conditions (*Li et al., 2015*). Previously, we reported a genome-wide study that H3pT11 specifically increases at the promoters of stress-responsive genes upon low glucose stress, and H3pT11 is required for proper transcription of those genes (*Oh et al., 2018*). Notably, the H3pT11 increase upon low glucose stress is a SESAME-independent process, indicating multiple different kinases are involved in H3pT11 regulation. We found that Cka1, a catalytic subunit of CK2 complex, is required for H3pT11 regulation in low glucose conditions. However, CK2 has been observed to be constitutively active and insensitive to external changes of environmental cues (*Pinna, 2002*). It remains elusive how a constantly active CK2 can regulate H3pT11, which becomes elevated upon low glucose stress. This suggests that H3pT11 may require additional mechanisms for sensing the availability of carbon sources.

In this study, we identify an understudied kinase, Tda1, as a histone H3 T11 kinase in low glucose stress conditions. Snf1 directly phosphorylates Tda1 at its C-terminus, especially at tandem serine 483 and threonine 484 (S483/T484) residues, which is required for in vivo Tda1 activity. CK2 regulates H3pT11 by controlling Tda1 nuclear localization. Hence, Tda1 acts as a signaling platform that can combine Snf1 and CK2 signaling, thereby connecting external nutrient availability to transcription regulation in the nucleus.

## Results

### H3pT11 upon low glucose stress is dependent on Snf1

We previously reported that H3pT11 can act as a marker of low glucose stress in yeast (*Oh et al., 2018*). The global H3pT11 level is inversely correlated with external glucose level and specifically increased at genes involved in metabolic changes. As AMPK signaling has major regulatory roles in nutrient starvation conditions (*González et al., 2020*), we tested whether H3pT11 is linked with the yeast AMPK homolog, Snf1 signaling pathway. Upon the media shift from glucose-rich (2%) YPD to nutritionally unfavorable YP with 3% glycerol (YPgly), Snf1 threonine 210 phosphorylation (Snf1 pT210), the active Snf1 marker, showed a similar pattern of increase compared to that of H3pT11 (*Figure 1A*). In addition, a *snf1Δ* mutant showed significantly reduced global H3pT11 levels in saturated cultures (*Figure 1B*) where media glucose is depleted (*Oh et al., 2018*) and impaired H3pT11 increase upon media shift from YPD to YPgly (*Figure 1C*). Rim15 is another kinase that becomes

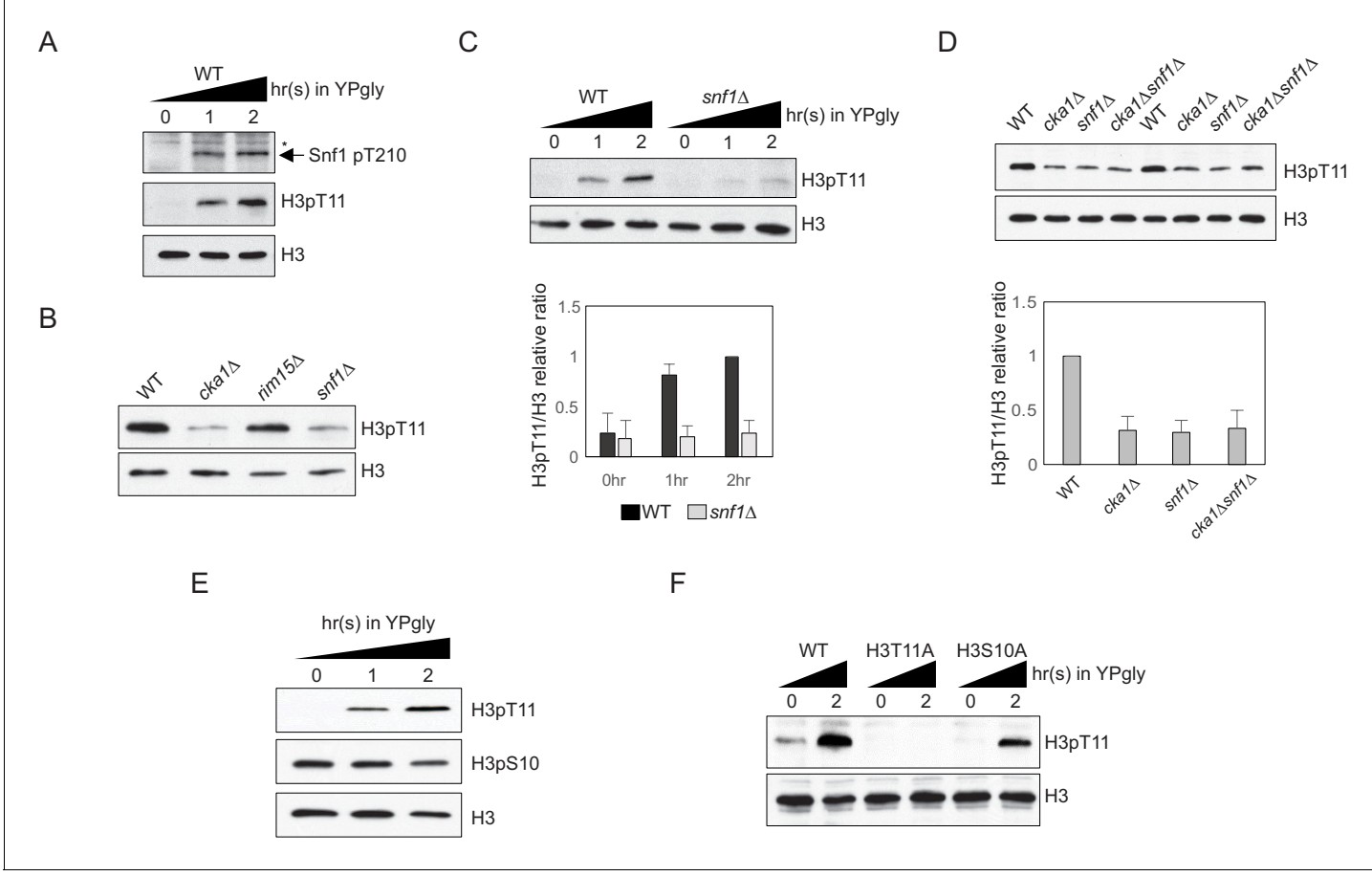

**Figure 1.** H3pT11 upon low glucose is dependent on Snf1, but not on H3pS10. (**A**) Comparison of the Snf1 threonine 210 phosphorylation (Snf1 pT210) and H3pT11 levels upon media shift from YPD (2% glucose) to YPgly (YP with 3% glycerol) measured by western blots. (**B**) Global H3pT11 levels in WT (BY4741), *cka1Δ*, *rim15Δ*, and *snf1Δ* cells measured by western blot. The cells were taken from overnight saturated cultures in YPD media. (**C**) (Upper panel) Comparison of H3pT11 levels in WT and *snf1Δ* upon the media shift from YPD to YPgly at indicated time points analyzed by western blots. (Lower panel) Relative ratios of H3pT11 to H3 signals presented with error bars indicating standard deviations (STD) of three biological replicates. (**D**) (Upper panel) Global H3pT11 levels in WT, *cka1Δ*, *snf1Δ*, and *cka1Δsnf1Δ* cells taken from saturated cultures in YPD media measured by western blots. (Lower panel) Relative band intensities of H3pT11 to H3 signals. Error bars indicate STD from three biological replicates. (**E**) Changes in H3pT11 and H3pS10 signals in the WT strain (BY4741) upon the media shift from YPD to YPgly at indicated time points measured by western blots. (**F**) Comparison of H3pT11 upon media shift from YPD to YPgly in WT (y1166), H3T11A, and H3S10A strains analyzed by western blots.

The online version of this article includes the following figure supplement(s) for figure 1:

**Figure supplement 1.** H3pT11 depends on Snf1, but not on Snf1 target transcription factors.

active in low glucose conditions (*Vidan and Mitchell, 1997*; *Wei et al., 2008*), but a *rim15Δ* mutant did not show the global H3pT11 defect (*Figure 1B*). Snf1 phosphorylates several transcription factors including Adr1, Cat8, Sip4, and Mig1 upon nutritional stress (*Kacherovsky et al., 2008*; *Lesage et al., 1996*; *Randez-Gil et al., 1997*; *Treitel et al., 1998*); however, deletion of these transcription factors did not affect H3pT11 levels (*Figure 1—figure supplement 1*). These results suggest that H3pT11 is specifically regulated via the Snf1 pathway and reduction of H3pT11 in a *snf1Δ* mutant is not an indirect result of a transcriptional defect. Previously, we showed that Cka1, the catalytic subunit of CK2, is required for the H3pT11 increase in low glucose stress conditions (*Oh et al., 2018*). Interestingly, a *cka1Δsnf1Δ* double deletion mutant showed a similar level of H3pT11 compared to a *snf1Δ* or *cka1Δ* single deletion mutant (*Figure 1D*), suggesting that CK2 and Snf1 may act in the same pathway for H3pT11 regulation.

Snf1 is required for H3pS10 at the *INO1* gene promoter in yeast (*Lo et al., 2001*). As H3 serine 10 is next to H3 T11, we tested whether histone H3 S10 phosphorylation (H3pS10) and H3pT11 behave similarly. Upon media shift from YPD to YPgly, the global H3pS10 level remained stable,

while H3pT11 was increased (*Figure 1E*). In addition, an H3S10A mutant, where histone H3 serine 10 is mutated into alanine, showed an unperturbed H3pT11 increase upon glucose depletion (*Figure 1F*). These results indicate that H3pT11 and H3pS10 are differently regulated under low glucose stress, and Snf1 signaling, but not H3pS10, is required for H3pT11 increase in the stress condition.

## Snf1 and CK2 are not direct kinases for H3pT11

Snf1 forms a functional AMPK complex with the regulatory β and γ subunits (*Ross et al., 2016*). When we measured H3pT11 levels from Snf1 complex subunit deletion mutants, interestingly, only the catalytic α subunit mutant, *snf1Δ*, showed a defect, while β (Gal83, Sip1, and Sip2) and γ subunit (Snf4) deletion mutants showed a comparable level of H3pT11 to WT (*Figure 2A*). Indeed, a β subunit mutant (*gal83Δ*) showed a slightly increased level of H3pT11. As Gal83 dictates Snf1 nuclear localization upon low glucose stress (*Vincent et al., 2001*), this result raised the possibility that Snf1 may not be a direct kinase or a major kinase for H3pT11 in vivo.

Our previous study showed that yeast TAP-purified CK2 can phosphorylate H3 at T11 in vitro (*Oh et al., 2018*). However, recombinant yeast Cka1 and recombinant human CK2 complex did not phosphorylate H3 at T11 in vitro, while both TAP purified and recombinant CK2 phosphorylated H3

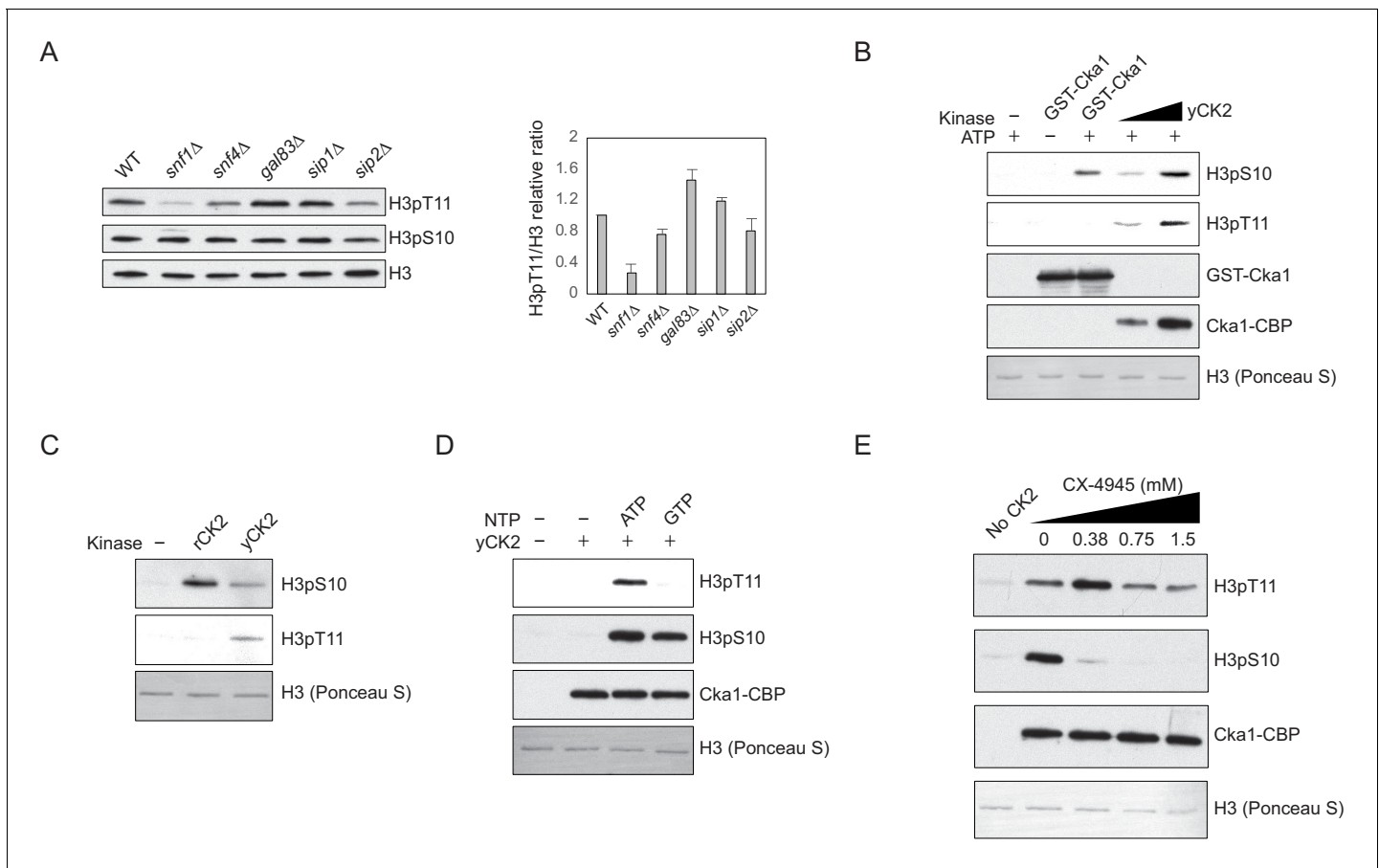

**Figure 2.** Snf1 and CK2 are not direct kinases for H3pT11. (A) (Left) Global H3pT11 and H3pS10 levels of Snf1 complex α subunit (Snf1), γ subunit (Snf4), and β subunit (Gal83, Sip1, and Sip2) deletion mutants compared to WT (BY4741) analyzed by western blots. Cells were taken from saturated cultures in YPD media. (Right) The relative band intensities of H3pT11 to H3 with error bars indicating STD of three biological replicates. (B) In vitro kinase assay of recombinant GST-Cka1 and yeast TAP purified CK2 (yCK2) using recombinant *Xenopus* histone H3 as a substrate. The reaction mixtures were incubated at 30°C for 3 hr. (C) In vitro kinase assay of human recombinant CK2 complex (rCK2) and yCK2 using recombinant H3 as a substrate. The reaction mixtures were incubated at 30°C for 1 hr. (D) In vitro kinase assay of yCK2 with recombinant H3 as a substrate and 5 mM ATP or GTP as a phosphate donor. All reaction mixtures were incubated at 30°C for 2 hr. (E) In vitro kinase assay of yCK2 for H3 with increasing amount of CX-4945 treatment. yCK2 was pre-incubated with CX-4945 at 30°C for 10 min, then mixed with H3 for additional 1 hr at 30°C.

at S10 (*Figure 2B,C*). CK2 is unusual for a kinase in that it can utilize both ATP and GTP as phosphate donors (*Niefind et al., 1999*). When GTP was used as the phosphate donor, TAP-purified CK2 phosphorylated H3 at S10, but not at T11 (*Figure 2D*). In addition, the CK2-specific inhibitor CX-4945 (*Siddiqui-Jain et al., 2010*) suppressed CK2 activity against H3pS10, but not against H3pT11 in vitro (*Figure 2E*). These results indicate that an unknown CK2-interacting kinase, rather than CK2 itself, is responsible for H3pT11 in vitro.

## Tda1 is responsible for H3pT11 upon low glucose stress in vivo

Since Snf1 and CK2 are not direct kinases against H3pT11, we inquired as to which kinase might be responsible for the modification. As the Snf1 pathway is critical in nutritional stress conditions, we hypothesized that Snf1-interacting kinases may be responsible or participate in H3pT11 regulation. Using the Saccharomyces Genome Database (https://www.yeastgenome.org) as a guide, we tested H3pT11 levels of deletion mutants of the kinases, which are known to interact with the Snf1 complex. Unexpectedly, the yeast Nuak1 homolog *tda1Δ* mutant showed a significantly reduced H3pT11 levels (*Figure 3A*). In glucose-depleted media, the *tda1Δ* mutant showed a more severe defect in the global H3pT11 levels than the *snf1Δ* mutant (*Figure 3B*). H3pS10 levels did not significantly change in *tda1Δ*, supporting the independence of H3pT11 from H3pS10 in this stress condition. In media-shift assays, the *tda1Δ* mutant showed impaired H3pT11 increase in YPgly, similar to *snf1Δ* (*Figure 3C*). Interestingly, upon media shift from YPD to YPgly, Tda1 protein levels were significantly increased in both the cytoplasm and the nucleus in the wild-type strain (*Figure 3D, Figure 3—figure*

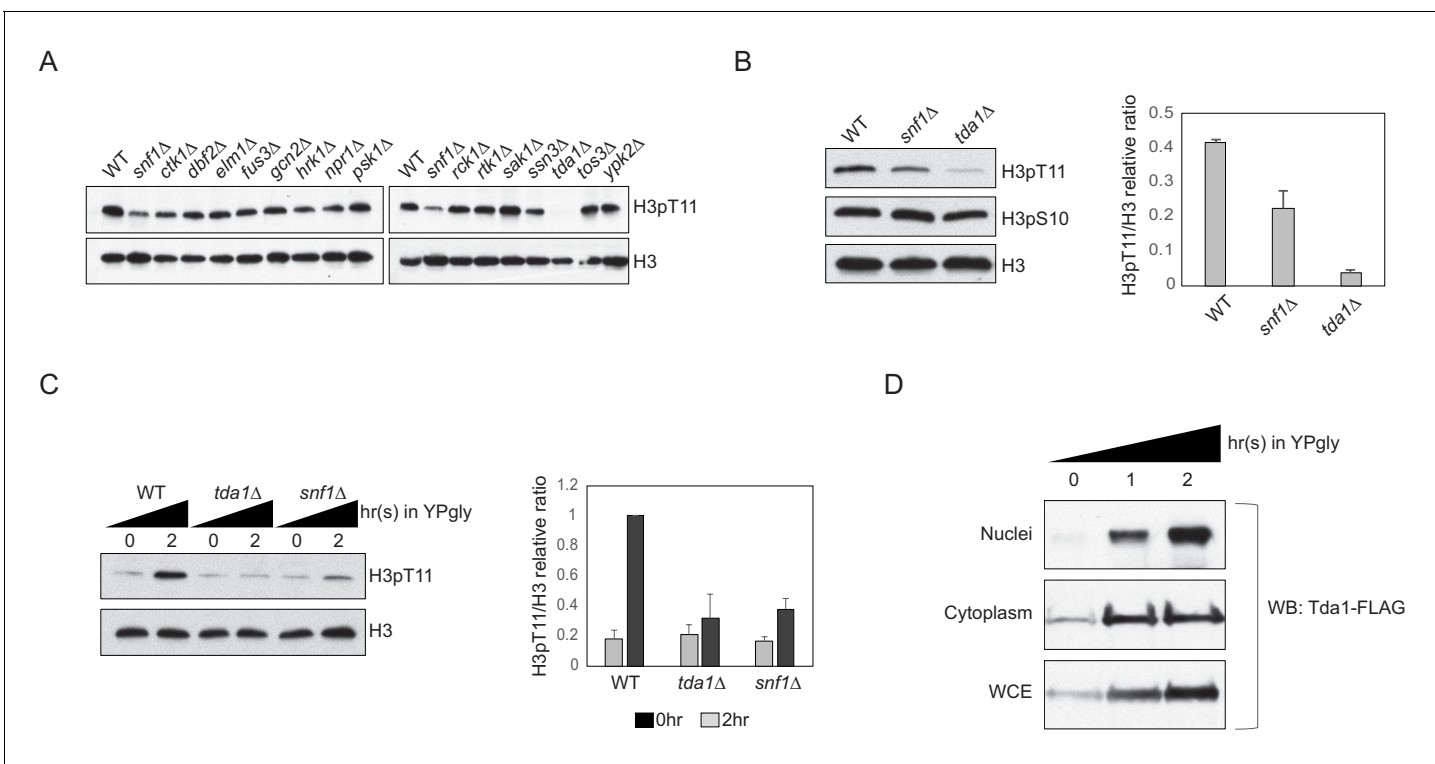

**Figure 3.** Tda1 is responsible for H3pT11 upon low glucose stress in vivo. (**A**) Global H3pT11 levels in Snf1-interacting kinase mutant cells taken from saturated cultures in YPD media measured by western blots. (**B and C**) (Left panels) H3pT11 levels in WT, *snf1Δ*, and *tda1Δ* cells taken from (**B**) saturated cultures in YPD media or taken at (**C**) indicated time points upon media shift from YPD to YPgly analyzed by western blots. (Right panels) The relative band intensities of H3pT11 to H3 are presented with error bars indicating STD of three biological replicates. (**D**) Tda1 protein levels tagged with C-terminal 3xFLAG tag in the nuclei, cytoplasm, and whole-cell extracts (WCE) upon media shift from YPD to YPgly media at indicated time points measured by western blots against FLAG tag.

The online version of this article includes the following figure supplement(s) for figure 3:

**Figure supplement 1.** Confirmation of efficient yeast subcellular fractionation of the samples shown in *Figure 3D*.

*supplement 1*). Collectively, these results strongly suggest that Tda1 has a central role in H3pT11 regulation upon low glucose stress.

## Tda1 phosphorylates H3 at T11 in vitro

Thus far, the only known Tda1 target is yeast Hxk2 at serine 15 (*Kaps et al., 2015*; *Kettner et al., 2012*). We noticed that the surrounding sequence of Hxk2 serine 15 is similar to that of histone H3 T11 (*Figure 4A*). To test whether Tda1 can directly phosphorylate H3 in vitro, we purified Tda1 from yeast using TAP tag purification (*Figure 4B*). TAP purified Tda1 did not show many bands in a poly-acrylamide gel electrophoresis (PAGE) gel, suggesting that Tda1 may not be a part of multi-subunit complex. An in vitro assay showed that indeed, TAP-purified Tda1 protein can robustly phosphory-late H3 at T11 (*Figure 4C*). Recombinant GST tagged Tda1 purified from *Escherichia coli* also phos-phorylated H3 at T11, but not at S10, while recombinant GST-tagged Cka1 phosphorylated H3 at S10 in vitro (*Figure 4D*). Sequence analysis from Saccharomyces Genome Database (https://www.yeastgenome.org/) predicts that Tda1 may contain a kinase domain at the N-terminus. In this regard, we made several Tda1 fragments (Tda1 N1–N5) to define the kinase domain (*Figure 4E*). An in vitro kinase assay of the fragments showed that Tda1 N1 (amino acids [aa] 1–353) did not phosphorylate H3 at T11 unlike Tda1 N2 (aa 1–380). Interestingly, Tda1 N3 (amino acids 1–426) showed a reduced activity against H3pT11 compared to Tda1 N2 (*Figure 4F*). These results suggest that Tda1 catalytic

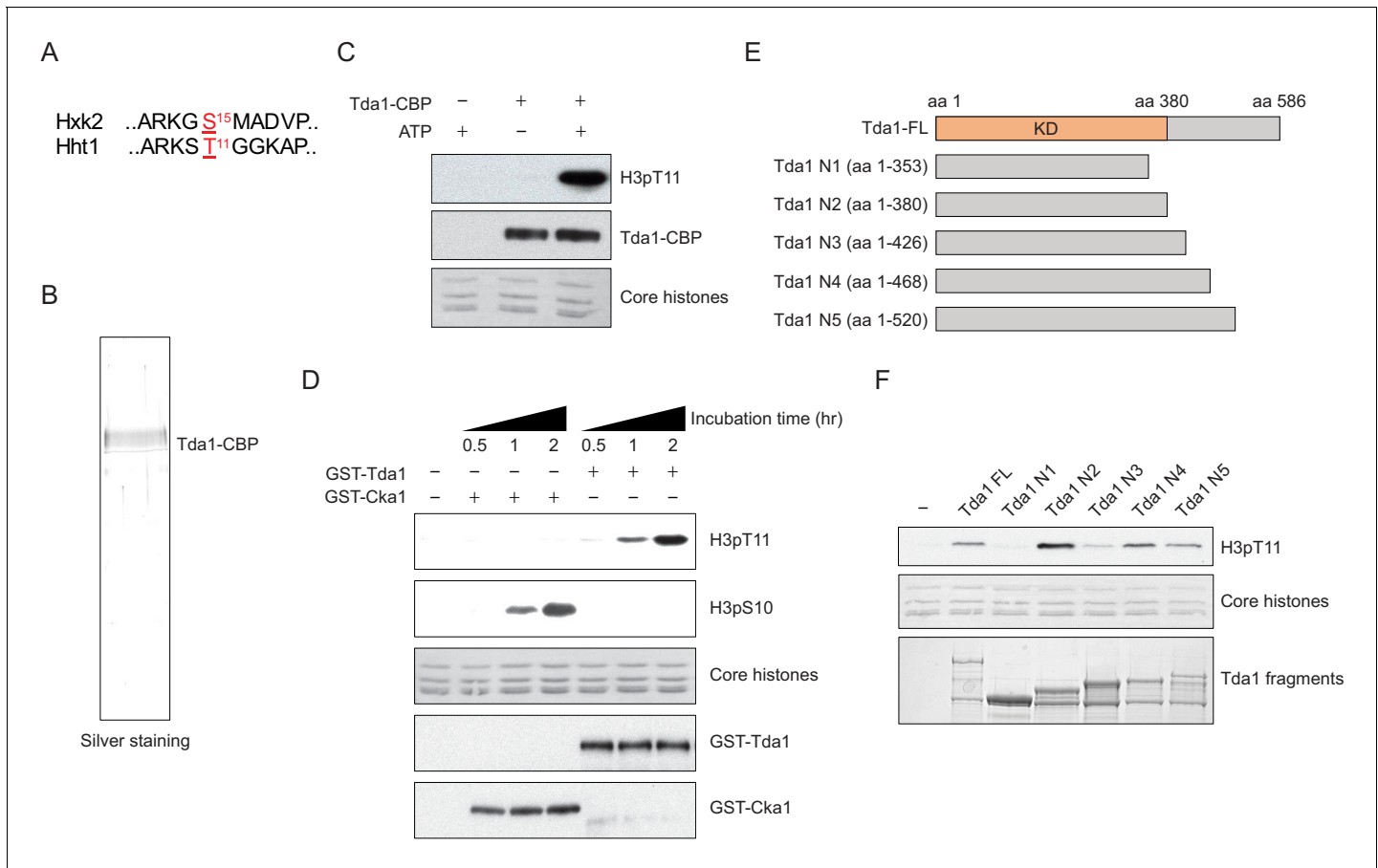

**Figure 4.** Tda1 phosphorylates H3 at T11 in vitro. (A) Comparison between the surrounding sequences of Hxk2 serine 15 and H3 (Hht1) threonine 11. (B) Silver staining of TAP-purified Tda1 protein. (C) In vitro kinase assay of TAP purified Tda1 using core histones (H2A, H2B, H3, and H4) as substrates. The reaction mixtures were incubated at 30°C for 1 hr. (D) In vitro kinase assay of recombinant GST-Cka1 and GST-Tda1 purified from *Escherichia coli* using core histones (H2A, H2B, H3, and H4) as substrates. The reaction mixtures were incubated at 30°C for indicated times. (E) Schematic diagram of recombinant GST-Tda1 N-terminal fragments used in (F). (F) In vitro kinase assay of Tda1 N fragments using core histones as substrates. All reaction mixtures were incubated at 30°C for 2 hr.

domain resides in aa 1–380, and the aa patch spanning 381–426 has a potential inhibitory effect on Tda1 activity against H3pT11.

## Snf1 directly phosphorylates Tda1 at C-terminus

Next, we inquired how Snf1 signaling can regulate the activity of Tda1 toward H3pT11. First, we compared the changes in Tda1 protein levels upon media shift from YPD to YPgly in WT and *snf1∆* (*Figure 5A*). Interestingly, Tda1 showed multiple species migrating differently on a PAGE gel. In a low glucose stress condition, slowly migrating Tda1 species became dominant in the WT strain. However, in the *snf1∆* mutant, the faster migrating species was dominant. In the *cka1∆* mutant, Tda1 protein-level changes were similar to that of WT (*Figure 5—figure supplement 1A*). We speculated that the apparent size differences of Tda1 in WT and *snf1∆* reflected differential post translational modification patterns of Tda1. To test this possibility, Tda1 was purified from yeast using FLAG tag, then incubated with λ phosphatase. After the phosphatase treatment, Tda1 showed a more distinctive one band pattern while the slower migrating bands disappeared, indicating Tda1 is indeed a phosphorylated protein (*Figure 5B*). In addition, when we compared Tda1 purified from WT, *snf1∆*, and *cka1∆* backgrounds, the phosphorylated species of Tda1 was significantly reduced specifically in the *snf1∆* mutant, implying that Snf1 is responsible for Tda1 phosphorylation (*Figure 5C*).

Tda1 co-immunoprecipitated with Snf1 as well as with Cka1 in vivo (*Figure 5—figure supplement 1B and ,C*). In this regard, we tested whether Snf1 can directly phosphorylate Tda1 in vitro. Snf1 was purified from a *reg1∆* strain using FLAG tag, maintaining a robust Snf1 pT210 signal (*Figure 5—figure supplement 1D*), as Reg1 recruits the Snf1 pT210 phosphatase Glc7 and inactivates Snf1 (*Tu and Carlson, 1995*). Then the purified Snf1-FLAG was incubated with GST-tagged Tda1 fragments, Tda1 N1 (aa 1–353) or Tda1C (aa 354–586) (*Figure 5D*). Interestingly, Snf1 robustly phosphorylated only the Tda1C fragment (*Figure 5E*). The pattern of protein band migration in the PAGE gel also indicated phosphorylated Tda1C. An in vitro assay with a recombinant Snf1 catalytic domain (aa 41–315, Snf1-CAT) also showed that Snf1 can phosphorylate Tda1C fragment (*Figure 5F*).

## Tda1 phosphorylation by Snf1 is required for Tda1 activity

To investigate which residues of Tda1 are modified by Snf1, the Tda1C fragment was incubated with recombinant Snf1-CAT or recombinant GST-Cka1 and then subjected to multi-dimensional protein identification technology (MudPIT) mass spectrometry analysis. Surprisingly, MudPIT analysis revealed that Snf1 phosphorylates multiple serine and threonine residues of the Tda1C fragment (*Figure 6A* and *Supplementary file 1*). We categorized Tda1 phosphorylation sites by Snf1 into three groups by their proximity to each other. Group I includes multiple serine and threonine residues residing in aa 396– 417. Interestingly, this region is located just after the Tda1 kinase domain. Group II includes tandem serine 483 and threonine 484. These two residues showed the highest PTM percentages among tested residues. Group III includes serine 570. MudPIT analysis also revealed that Cka1 robustly phosphorylated Tda1C at serine 578.

To understand how Tda1 phosphorylation by Snf1 or Cka1 regulates its function, we generated Tda1 phosphorylation site-defective mutants and tested their H3pT11 levels. Strikingly, among the tested mutants, H3pT11 level was significantly reduced only in the group II site-defective mutant, Tda1 S483A/T484A (*Figure 6B*). Tda1 S483A/T484A mutant also showed impaired H3pT11 increase, while Tda1 ∆396–417 mutant showed intact H3pT11 increase upon media shift from YPD to YPgly (*Figure 6C*). Interestingly, Both Tda1 S483A/T484A and ∆396–417 mutants showed the robust increase of Tda1 phosphorylation upon media shift (*Figure 6C* Tda1-FLAG panel), which was significantly reduced in *snf1∆* background (*Figure 5A*). These results suggest that Snf1 is responsible for the multi-sites phosphorylation of Tda1 and among those phosphorylation, S483/T484 phosphorylation has a critical role for H3pT11 regulation.

We inquired how Tda1 S483/T484 phosphorylation can regulate the function of Tda1 for H3pT11. First, we hypothesized that the availability of Tda1 for H3pT11 may be affected by its phosphorylation. However, Tda1 S483A/T484A mutant showed a similar level of nuclear Tda1 compared to Tda1 WT (*Figure 6—figure supplement 1A,B*), suggesting that Tda1 S483/T484 phosphorylation is not required for Tda1 nuclear localization. Next, we hypothesized that Tda1 S483/T484 phosphorylation may regulate Tda1 activity against H3T11. To test this hypothesis, we purified Tda1 WT or Tda1

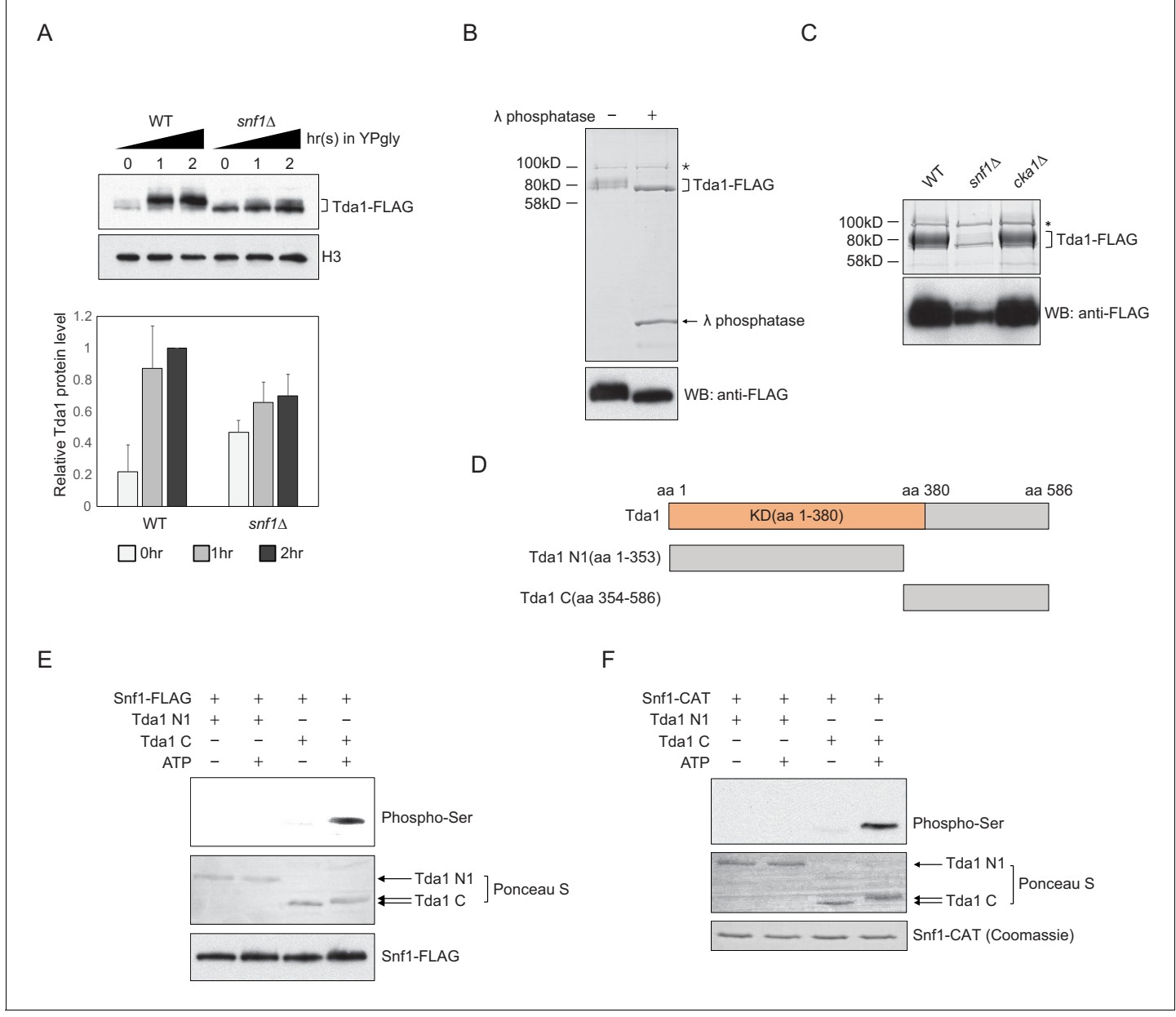

**Figure 5.** Snf1 phosphorylates Tda1 at C-terminus. (**A**) (Upper panel) Tda1-3xFLAG protein level changes in WT and *snf1Δ* upon media shift from YPD to YPgly measured by western blot against FLAG tag. (Lower panel) Relative band intensities of Tda1 to H3 with error bars indicating STD of three biological replicates. (**B**) Coomassie staining (upper panel) or western blot (WB) against FLAG tag (lower panel) of yeast FLAG purified Tda1 with or without λ phosphatase treatment. (**C**) Coomassie staining (upper panel) or western blot against FLAG tag (lower panel) of yeast FLAG purified Tda1 in WT, *snf1Δ*, and *cka1Δ* background. (**D**) Schematic diagram of Tda1 N1 (Tda1 aa 1–353) and Tda1 C (Tda1 aa 354–586) used for in vitro kinase assays shown in (**E**) and (**F**). (**E**) In vitro kinase assay of yeast FLAG purified Snf1 from *reg1Δ* background using GST-Tda1 N1 or GST-Tda1 C as a substrate. The reaction mixtures were incubated for 2 hr at 30°C. (**F**) In vitro kinase assay of recombinant Snf1 catalytic domain (Snf1-CAT) which was activated by human CaMKK2 using GST-Tda1 N1 or GST-Tda1 C as a substrate. The reaction mixtures were incubated at 30°C for 2 hr.

The online version of this article includes the following figure supplement(s) for figure 5:

**Figure supplement 1.** Snf1 phosphorylates Tda1 at C-terminus.

S483A/T484A protein from yeast for in vitro kinase assays. Interestingly, the Tda1 S483A/T484A mutant showed significantly lower activity against H3 T11 than Tda1 WT (*Figure 6D*). These results indicate that Snf1 regulates Tda1 activity by phosphorylating Tda1 S483/T484 residues.

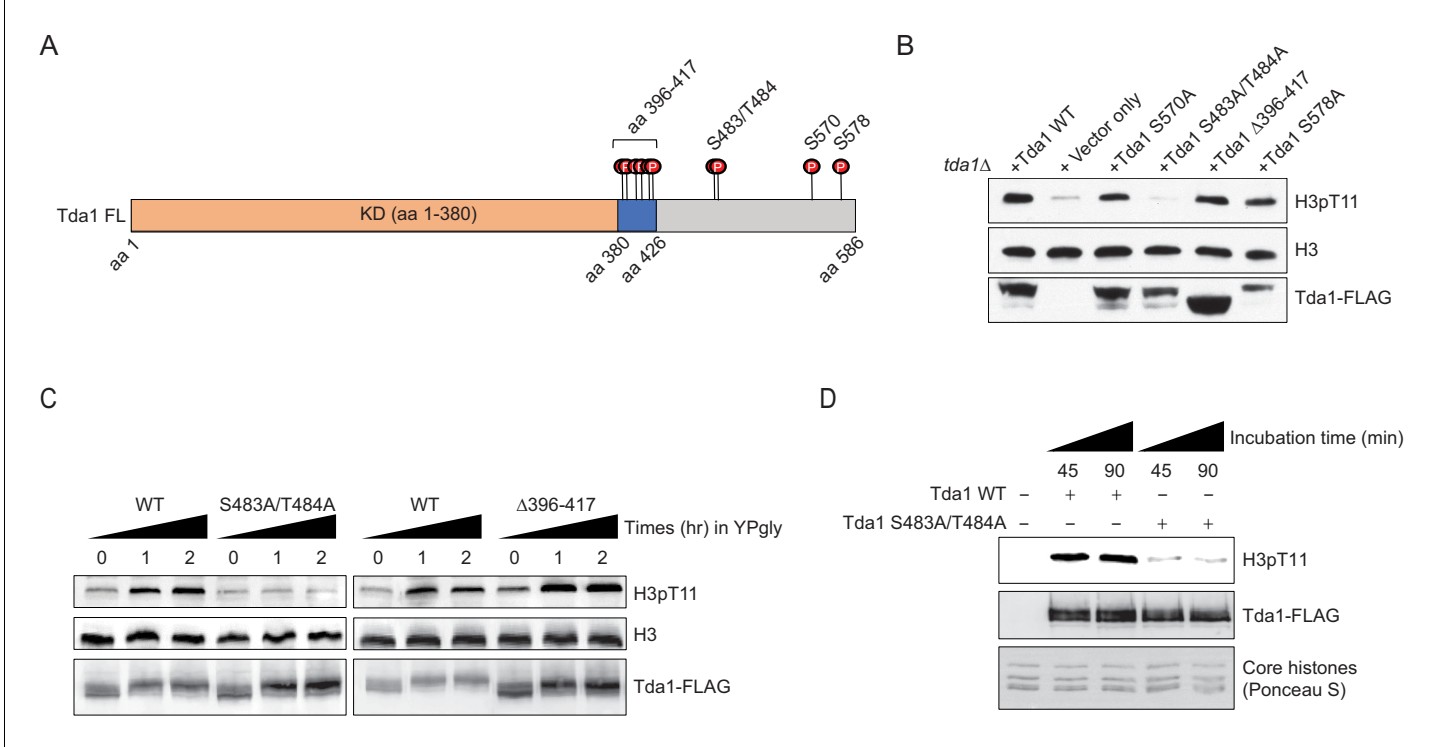

**Figure 6.** Tda1 S483/T484 phosphorylation by Snf1 is required for the Tda1 activity in vivo. (**A**) Schematic map of Tda1 phosphorylation sites (denoted as red circles) by recombinant Snf1-CAT and Cka1. (**B**) Global H3pT11 levels of Tda1 phosphorylation sites defective mutants taken from saturated cultures in YPD media. The Tda1 constructs were expressed in pRS416 shuttling vectors and their expression was governed by ADH1 promoter. 'Vector only' construct indicates the pRS416 with ADH1 promoter only. (**C**) H3pT11 levels upon media shift from YPD to YPgly in *tda1Δ* cells expressing Tda1 WT, Tda1 S483A/T484A, or Tda1 Δ396–417 constructs under an ADH1 promoter. (**D**) In vitro kinase assay of yeast FLAG purified Tda1 WT or Tda1 S483A/T484A protein using core histones as substrates. All reaction mixtures were incubated at 30°C for indicated times.

The online version of this article includes the following source data and figure supplement(s) for figure 6:

**Figure supplement 1.** Tda1 S483/T484 phosphorylation by Snf1 is required for the nuclear Tda1 activity.

**Figure supplement 2.** Tda1 S483/T484 phosphorylation is required for the transcription of H3pT11 target genes.

**Figure supplement 2—source data 1.** Raw data used for *Figure 6—figure supplement 2*.

As H3pT11 is required for the transcription of stress-responsive genes upon low glucose stress (*Oh et al., 2018*), we asked whether the transcription of H3pT11 target genes was affected by Tda1 S483/T484 phosphorylation. By using RNA-Seq results published in our previous work (*Oh et al., 2018*), we selected four known H3pT11 target genes: *HSP26*, *GRE1*, *GND2*, and *SOL4*, and two non-target genes: *MET13* and *PMA1*. The expression levels of H3pT11 target genes were significantly increased upon media shift from YPD to YPgly (*Figure 6—figure supplement 2A*) and impaired in the H3T11A mutant (*Figure 6—figure supplement 2B*). Interestingly, the Tda1 S483A/T484A (AA) mutant could not restore the transcription of H3pT11 target genes and show a similar transcription level to *tda1Δ* (Δ) mutant (*Figure 6—figure supplement 2C*). At non-target genes, WT, *tda1Δ*, and AA mutant showed a similar level of gene expression (*Figure 6—figure supplement 2D*). These results imply that Tda1 specifically regulates its target gene transcription via H3pT11 phosphorylation, and Tda1 S483/T484 phosphorylation by Snf1 has critical roles in the regulation of Tda1 activity.

## CK2 regulates Tda1 nuclear localization

Both Snf1 and CK2 are required for proper H3pT11 levels in low glucose stress conditions, and they genetically interact with each other in H3pT11 regulation (*Figure 1D*). CK2 does not affect global Tda1 phosphorylation levels (*Figure 5C*). Tda1 purified from *cka1Δ* background showed similar activity against H3 T11 to Tda1 purified from WT (*Figure 7—figure supplement 1*), suggesting that CK2 does not affect Tda1 activity. To investigate how CK2 regulates Tda1 function, we tested Tda1

subcellular localization in WT and *cka1Δ* mutant. Interestingly, we found that Tda1 nuclear localization was significantly decreased in *cka1Δ* mutants in low glucose stress conditions compared to WT (*Figure 7A*). As we could not find any conventional NLS sequence in Tda1, we attached a strong cMyc nuclear localization signal (PAAKRVKLD) (*Dang and Lee, 1988*) at the C-terminal end of Tda1 to see whether the cMyc NLS could help Tda1 to bypass the requirement for Cka1 for its nuclear localization. The DNA construct expressing Tda1 tagged with C-terminal 3xFLAG (Tda1 WT) or 3xFLAG with cMyc NLS (Tda1 cNLS) under a strong ADH1 promoter was integrated into the genome of a *tda1Δ* mutant. Upon low glucose stress, Tda1 cNLS showed a slightly lower level of the protein increase than Tda1 WT. However, the Tda1 cNLS expressing strain showed a more robust and rapid increase of H3pT11 compared to the Tda1 WT expressing strain (*Figure 7B*). Next, we integrated the Tda1 WT or Tda1 cNLS construct into *tda1Δcka1Δ* or *tda1Δsnf1Δ* backgrounds. Interestingly, in the *cka1Δ* mutant, the Tda1 cNLS was able to mediate H3pT11, unlike Tda1 WT (*Figure 7C*, left panel). On the contrary, neither Tda1 cNLS nor Tda1 WT could restore H3pT11 in *snf1Δ* mutant (*Figure 7C*, right panel). Thus, while Snf1 regulates Tda1 activity, Cka1 regulates its nuclear localization. These results clearly show that Snf1 and Cka1 signaling pathways cooperatively regulate H3pT11 but use different mechanisms or act in different stages of the Tda1 regulation (summarized in *Figure 7D*).

## Discussion

### Tda1 phosphorylation by Snf1

In vitro kinase assays using recombinant Tda1 fragments and Snf1 revealed multiple Tda1 residues phosphorylated by Snf1 (*Figure 6A* and *Supplementary file 1*). Among Tda1 phosphorylation site-defective mutants tested, only a Tda1 S483A/T484A mutant showed a significant defect in H3pT11 regulation in vivo (*Figure 6B,C*), suggesting the importance of Tda1 S483/T484 phosphorylation. Indeed, the Tda1 S483/T484 residues showed the highest level of modification in the in vitro kinase assay (*Supplementary file 1*).

Tda1 S483/T484 phosphorylation is required for Tda1 kinase activity on H3 T11 (*Figure 6D*); however, how the modification regulates Tda1 activity remains unclear. Tda1 S483/T484 residues are not located in the Tda1 catalytic domain (*Figure 4E*, Tda1 aa 1–380). Tda1 S483/T484 phosphorylation may regulate Tda1 activity by relieving an unknown inhibition mechanism, or Tda1 may require additional factors to become fully active, and Tda1 phosphorylation at S483/T484 may be responsible for interacting with such factors. Investigating Tda1 binding proteins which depend on Tda1 phosphorylation would be an exciting future step.

### Cka1 regulates Tda1 nuclear localization

Both Snf1 and Cka1 are required for H3pT11 regulation, but two kinases regulate H3pT11 via different mechanisms. Snf1 robustly phosphorylates the Tda1 C-terminal (*Figure 5E,F*), while Cka1 does not affect global Tda1 phosphorylation (*Figure 5C*). A Tda1 S578A point mutation at the putative phosphorylation site by Cka1 does not affect the global level of H3pT11 (*Figure 6B*). Tda1 with cMyc NLS behaves differently in the *snf1Δ* and *cka1Δ* backgrounds (*Figure 7C*). The NLS-attached Tda1 can bypass the requirement for Cka1, but not that for Snf1. This result suggests that Cka1 regulates the nuclear localization process of Tda1 by controlling the interaction between Tda1 and nuclear importins, in contrast to Snf1 that regulates Tda1 enzymatic activity (*Figure 6D*). Tda1 does not possess conventional phosphorylation NLS sequence in it. In this regard, it is not clear if Tda1 itself contains non-conventional NLS, or Tda1 requires other accessory factors for its nuclear localization. Finding a Tda1 specific importin would also be an important next step.

CK2 has been known as a constitutively active kinase complex, which does not require upstream stimulation for its activity. However, many processes governed by CK2 are highly controlled processes (*Pinna, 2002*). In this study, we show an example of how a constitutively active kinase can be involved in tightly regulated processes such as H3pT11 upon low glucose stress. Although CK2 is constitutively active, H3pT11 cannot be achieved without active Snf1. Previously, we reported that Sch9 was also required for H3pT11 regulation as well as CK2 (*Oh et al., 2018*). Interestingly, Sch9 and CK2 genetically interact in H3pT11 regulation as Cka1 and Snf1 do. Our finding of H3pT11 regulation by CK2 via controlling the Tda1 nuclear localization suggests a possibility that Sch9 is also

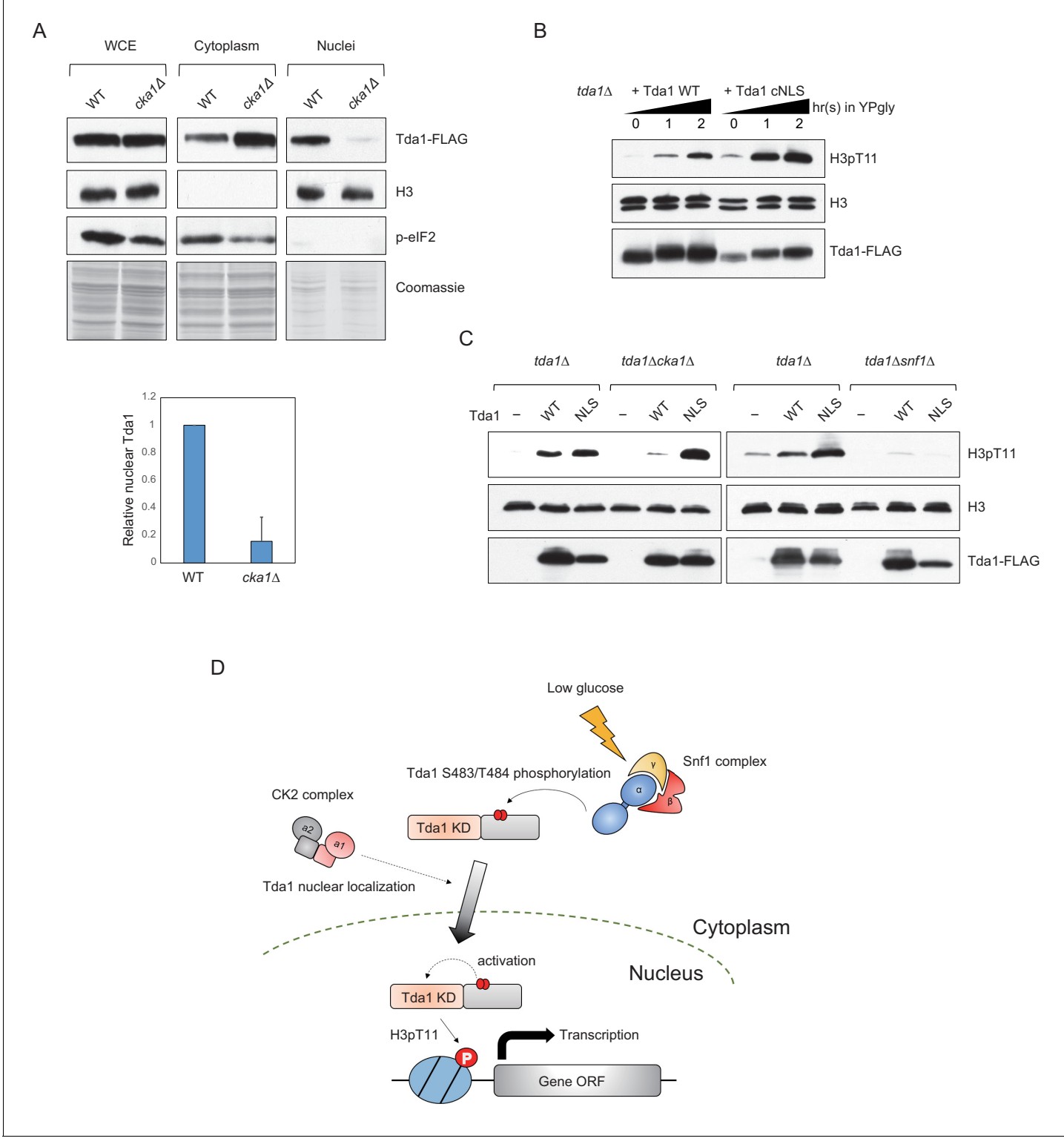

**Figure 7.** CK2 regulates Tda1 nuclear localization. (**A**) (Upper panels) Subcellular localization of Tda1-3xFLAG proteins in WT and *cka1Δ* mutant. whole-cell extract (WCE), cytoplasm, and nuclei samples were taken from YPgly media cultures incubated for 2 hr after the media shift from YPD, then analyzed by western blot. Histone H3 and p-eIF2 antibodies were used for the nuclear and cytoplasmic marker, respectively. (Lower panel) The relative band intensities of nuclear Tda1 to WCE Tda1 are presented with error bars indicating STD of three biological replicates. (**B**) H3pT11-level changes in Tda1 WT and Tda1 with C-terminally tagged cMyc NLS upon media shift from YPD to YPgly analyzed by western blots. (**C**) H3pT11 restoration by genome integrated Tda1 WT or Tda1 with cMyc NLS in *tda1Δ*, *tda1Δcka1Δ*, and *tda1Δsnf1Δ* backgrounds measured by western blots. The cells were

*Figure 7 continued on next page*

*Figure 7 continued*

taken from YPgly media cultures incubated for 2 hr after the media shift from YPD. (D) Summary model of the Tda1 regulation by Snf1 and Cka1. The red ellipses represent phosphorylation.

The online version of this article includes the following figure supplement(s) for figure 7:

**Figure supplement 1.** CK2 does not regulate Tda1 activity.

involved in the same process. This raises the possibility of Tda1 being a signaling platform that combines multiple signaling such as Snf1, CK2, and Sch9 together to finely connect external glucose levels to chromatin.

## Tda1 in higher eukaryotes

Structural analysis (at http://yeastgenome.org) predicts that Tda1 contains a calcium/calmodulin-dependent kinase domain at its N-terminus. We found that its functional kinase domain resides between aa 1 and 380 (*Figure 4F*). In yeast, there are 12 kinases that contain the domain including yeast calcium/calmodulin-dependent kinases (Cmk1 and Cmk2) and meiosis-specific kinase Mek1, as well as Tda1. Interestingly, Mek1 phosphorylates H3 at T11 during meiosis (*Kniewel et al., 2017*). A large-scale screening of yeast phosphorylation site motifs revealed that the aa sequence surrounding H3T11 fits in the recognition motif of Cmk1 and Cmk2 (*Mok et al., 2010*). These results imply that calcium/calmodulin-dependent kinases could be kinase candidates for H3pT11 in situations other than nutrient starvation in yeast and possibly in other species.

Based on predictions by the PPOD program (at http://ppod.princeton.edu/), Nuak1 has been proposed as a human ortholog of yeast Tda1 (*Soma et al., 2014*). Nuak1 is also known as ARK5, which means AMPK-related kinase 5. As its name suggests, Nuak1 has a 47% aa sequence homology to AMPK α1 (*Suzuki et al., 2003a*). Nuak1 is involved in cell survival processes upon nutrient starvation (*Suzuki et al., 2003b*), suggesting that the Tda1 function upon nutrient stress shown in yeast is consistent in higher eukaryotes. Nuak1 predominantly localizes to the nucleus, but a recent study revealed that the cellular distribution of Nuak1 changes upon stresses in a importin β-dependent manner (*Palma et al., 2019*). Investigating whether the Nuak1 subcellular localization is regulated via a similar mechanism to that of Tda1 and phosphorylates histones under a specific stress like Tda1 would be an interesting future study.

## Materials and methods

### Key resources table

| Reagent type (species) or resource | Designation | Source or reference | Identifiers | Additional information |
|---|---|---|---|---|
| Strain, strain background (*Saccharomyces cerevisiae*) | BY4741 | Open Biosystems | Cat# YSC1048 | All yeast strains derived from BY4741 are listed in *Supplementary file 2* |
| Strain, strain background (*Saccharomyces cerevisiae*) | W303-1A | Euroscarf | Cat# BMA64-1A | All yeast strains derived from W303-1A are listed in *Supplementary file 2* |
| Antibody | Anti-H3pT11 (Rabbit polyclonal) | Abcam | Cat# ab5168, RRID: AB_304759 | (1:100) |
| Antibody | Anti-H3 (Rabbit polyclonal) | Abcam | Cat# ab1791, RRID: AB_302613 | (1:1000) |
| Antibody | Anti-FLAG (Mouse monoclonal) | MilliporeSigma | Cat# F1804, RRID: AB_262044 | (1:10,000) |
| Antibody | Anti-HA (Rat monoclonal) | Roche | Cat# 12013819001, RRID:AB_390917 | (1:1000) |

*Continued on next page*

*Continued*

| Reagent type (species) or resource | Designation | Source or reference | Identifiers | Additional information |
|---|---|---|---|---|
| Antibody | Anti-Snf1 pT210 (Rabbit polyclonal) | Cell Signaling | Cat# 2531, RRID: AB_330330 | (1:100) |
| Antibody | Anti-H3pS10 (Rabbit polyclonal) | Abcam | Cat# ab5176, RRID: AB_304763 | (1:1000) |
| Antibody | Anti-GST (Rabbit polyclonal) | Santa Cruz Biotechnology | Cat# sc-459, RRID: AB_631586 | (1:500) |
| Antibody | Anti-phosphoserine (Rabbit polyclonal) | MilliporeSigma | Cat# AB1603, RRID: AB_390205 | (1:1000) |
| Antibody | Anti-phospho-eIF2alpha (Rabbit polyclonal) | Cell signaling | Cat# 9721, RRID: AB_330951 | (1:500) |
| Antibody | Anti-CBP (Rabbit polyclonal) | *Venkatesh et al., 2012*, PMID:22914091 | N/A | (1:2000) |
| Recombinant DNA reagent | pET29a-YS14 | Addgene | Cat# 66890 RRID:Addgene_66890 | |
| Recombinant DNA reagent | pFA6A-NatMX6 | Euroscarf | Cat# P30437 | |
| Recombinant DNA reagent | pFA6a–6XGLY–3XFLAG–HIS3M × 6 | Addgene | Cat# 20753 RRID:Addgene_20753 | |
| Recombinant DNA reagent | pFA6A-HIS3M × 6 | Addgene | Cat# 41596 RRID:Addgene_41596 | |
| Recombinant DNA reagent | pGEX4T-1 | MilliporeSigma | Cat# GE28-9549-49 | |
| Recombinant DNA reagent | pGEX4T2-Snf1-cat | Addgene | Cat# 52683 RRID:Addgene_52683 | |
| Recombinant DNA reagent | pRS416-ADH1 promoter | In this study, *Figure 6B* | N/A | |
| Recombinant DNA reagent | pRS416-ADH1 promoter-Tda1 WT 3xFLAG | In this study, *Figure 6B* | N/A | |
| Recombinant DNA reagent | pRS416-ADH1 promoter-Tda1 Δ396–417 3xFLAG | In this study, *Figure 6B* | N/A | |
| Recombinant DNA reagent | pRS416-ADH1 promoter-Tda1 S483A/T484A 3xFLAG | In this study, *Figure 6B* | N/A | |
| Recombinant DNA reagent | pRS416-ADH1 promoter-Tda1 S570A 3xFLAG | In this study, *Figure 6B* | N/A | |
| Recombinant DNA reagent | pRS416-ADH1 promoter-Tda1 S578A 3xFLAG | In this study, *Figure 6B* | N/A | |
| Recombinant DNA reagent | pRS406-ADH1 promoter-Tda1 WT 3xFLAG | In this study, *Figure 6C* | N/A | |
| Recombinant DNA reagent | pRS406-ADH1promoter-Tda1 Δ396–417 3xFLAG | In this study, *Figure 6C* | N/A | |
| Recombinant DNA reagent | pRS406-ADH1promoter-Tda1 S483A/T484A 3xFLAG | In this study, *Figure 6C* | N/A | |
| Recombinant DNA reagent | pRS406-ADH1promoter-Tda1 WT 3xFLAG cNLS | In this study, *Figure 7C* | N/A | |
| Peptide, recombinant protein | Casein Kinase II | New England Biolabs | Cat# P6010 | |

*Continued on next page*

*Continued*

| Reagent type (species) or resource | Designation | Source or reference | Identifiers | Additional information |
|---|---|---|---|---|
| Peptide, recombinant protein | CaMKK2 | Abnova | Cat# H00010645-P01 | |
| Peptide, recombinant protein | Lambda phosphatase | New England Biolabs | Cat# P0753 | |
| Peptide, recombinant protein | rLys-C, Mass Spec Grade | Promega | Cat# V1671 | |
| Peptide, recombinant protein | Sequencing Grade Modified Trypsin | Promega | Cat# V5111 | |
| Commercial assay or kit | DNA-free DNA Removal Kit | ThermoFisher | Cat# AM1906 | |
| Commercial assay or kit | ImProm II reverse transcription system | Promega | Cat# A3800 | |
| Chemical compound, drug | CX-4945 | Abcam | Cat# ab141350 | |
| Chemical compound, drug | Glycerol | Fisher Scientific | Cat# BP229 | |
| Chemical compound, drug | ATP | Thermo Scientific | Cat# R0441 | |
| Chemical compound, drug | GTP | Thermo Scientific | Cat# R0461 | |
| Software, algorithm | ProLuCID | *Eng et al., 1994*, PMID:24226387 | http://fields.scripps.edu/yates/wp/ | |
| Software, algorithm | DTAselect | *Tabb et al., 2002*, PMID:12643522 | http://fields.scripps.edu/yates/wp/ | |
| Software, algorithm | CONTRAST | *Tabb et al., 2002*, PMID:12643522 | http://fields.scripps.edu/yates/wp/ | |

## Yeast strains and culture conditions

All yeast strains used in this study are listed in *Supplementary file 2*. All single-deletion mutants using KanMX4 cassette and TAP-tagged strains derived from BY4741 were obtained from Open Biosystems library (maintained at the Stowers Institute Molecular Biology facility). Yeast synthetic histone H3 mutants (*Dai et al., 2008*) (H3 WT, H3 T11A, and H3 S10A in *Figure 1F*) were also purchased from Open Biosystems. Other deletions and tagged strains were made by targeted homologous recombination of PCR fragments containing marker genes flanked by gene specific sequences. These strains were confirmed by PCR with primer sets specific for their marker genes. For Tda1 mutant strains shown in *Figures 6C,D* and *7*, Tda1 WT or Tda1 mutant constructs were cloned into pRS406 vector, and the DNA constructs were linearized by NcoI digestion and then integrated into URA3 loci of the genome. For low glucose stress experiments, yeast cells were saturated by overnight culture at 30˚C. For media shift experiments, the saturated cultures were inoculated into fresh YPD media and then incubated until mid-log phase (optical density [OD] 0.4–0.6). These cultures were washed once with YPgly (YP with 3% glycerol) media, then resuspended with YPgly for indicated times at 30˚C.

## Preparation of yeast whole-cell extracts

Yeast whole-cell extracts were prepared as previously described (*Oh et al., 2018*). Five OD cells were taken from the cultures. Cell pellets were washed once with distilled water, then resuspended in 250 µL of 2M NaOH with 8% β-mercaptoethanol for 5 min on ice. Cells were pelleted then washed once with 250 µL TAP extraction buffer (40 mM HEPES pH 7.5, 10% glycerol, 350 mM NaCl, 0.1% Tween-20, phosphatase inhibitor cocktail, and proteinase inhibitor cocktail from Roche). Cell pellets

were resuspended in 180 µL modified 2× sodium dodecyl sulfate (SDS) buffer then boiled at 100°C for 4–5 min. For detecting Snf1 T210 phosphorylation (*Figure 1A*), cell cultures were boiled for 5 min before initial harvesting to prevent Snf1 phosphorylation by centrifugation (*Orlova et al., 2008*).

## Yeast flag-tagged protein purification

Tda1 and Snf1 proteins with 3xFLAG tag were purified as previously described (*Dutta et al., 2014*) with minor modifications. Six liter cultures of cells were grown in YPD to 2–3 OD at 30°C and collected. For Tda1-3xFLAG purification, cell pellets were resuspended in 3L YP without glucose and then incubated for an additional 1 hr. Cell pellets were washed once with distilled water, then resuspended in buffer A (25 mM HEPES pH 7.5, 10% glycerol, 350 mM KCl, 2 mM MgCl$_2$, 1 mM ethylenediaminetetraacetic acid [EDTA], 0.02% NP40, supplemented with 20 µg/mL leupeptin, 20 µg/mL pepstatin, and 100 µM phenylmethylsulfonyl fluoride [PMSF]) and then broken up by bead beating. The crude cell extracts were incubated with 125U benzonase and 500 µg heparin for 15 min at RT to remove nucleic acid contamination, then the extracts were further clarified by ultracentrifugation. The clarified extracts were incubated with anti-FLAG M2 affinity resin (Sigma) for 4 hr at 4°C with gentle rotation. Proteins bound to the resin were washed three times with buffer A, then washed once with buffer B (25 mM HEPES pH 7.5, 10% glycerol, 100 mM KCl, 2 mM MgCl$_2$, 1 mM EDTA, 1 mM dithiothreitol [DTT], 0.02% NP40, supplemented with 20 µg/mL leupeptin, 20 µg/mL pepstatin, and 100 µM PMSF). Bound proteins were eluted in buffer B containing 0.5 mg/mL 3xFLAG peptides.

## Recombinant protein purification

Yeast Cka1 and Tda1 genes were amplified by PCR from yeast genomic DNA and then cloned into pGEX4T-1 (GE Healthcare) vector. GST-Snf1 catalytic domain (Snf1-CAT) expression vector was purchased from Addgene (#52683). Those DNA constructs were transformed in Rosetta2 (Novagen) competent cells, and protein expressions were induced by 0.5 mM IPTG for 18 hr at 16°C. Bacterial cell pellets were resuspended in TAP extraction buffer, then sonicated to disrupt cell walls. Crude extracts were incubated with glutathione sepharose 4B (GE Healthcare) resin for 3 hr at 4°C. Resin-bound proteins were eluted in 50 mM Tris pH 8.0 buffer containing 20 mM glutathione. *Xenopus* core histones were purified as previously described (*Shim et al., 2012*) with minor modifications. Briefly, YS-14 construct encoding all four *Xenopus* core histones (H2A, H2B, H3, and H4) was transformed into Tuner DE3 pLysS competent cells (Novagen), and then histone protein expressions were induced in 2X YT media by 0.5 mM IPTG for 24 hr at 37°C. Cell pellets were sonicated in high salt extraction buffer (20 mM Tris pH 8.0, 2M NaCl), and then clarified extracts were incubated with TALON metal affinity resin (Clontech) for 3 hr at 4°C. Resin-bound proteins were eluted by 250 mM imidazole, and then eluted proteins were dialyzed in high salt extraction buffer for overnight at 4°C.

## Yeast subcellular fractionation

Yeast cellular fractionation was carried out as previously described (*Keogh et al., 2006*) with minor modifications. Forty to 50 OD yeast cells were pelleted, then washed successively with 10 mL distilled water and ice cold 10 mL SB (1 M sorbitol, 20 mM Tris–Cl pH 7.5), The washed cell pellets were transferred to 1.7 mL Eppendorf tube, then successively washed with 1 mL PSB (20 mM Tris–Cl pH 7.5, 2 mM EDTA, 100 mM NaCl, 10 mM β-mercaptoethanol) and 1 mL SB. The pellets were resuspended with 750 µL SB, then yeast cell walls were digested by adding 100 µL Zymolyase (10 mg/mL, Seikagaku) for 1 hr at RT. After the cell wall digestion, 750 µL ice-cold SB was added, then the spheroplasts were collected by gentle centrifugation (2k, 5 min, 4°C) and washed once with 1 mL ice-cold SB. The spheroplasts were resuspended by 500 µL EBX (20 mM Tris–Cl pH 7.5, 100 mM NaCl, 0.25% Triton X-100, 15 mM β-mercaptoethanol), then 100% Triton X-100 was added up to 0.5% to disrupt outer cell membrane. Cells were placed on ice for 10 min with occasional mixing, then 1 mL NIB (20 mM Tris–Cl pH 8.0, 100 mM NaCl, 1.2 M sucrose, 15 mM β-mercaptoethanol) was layered over the cells. After high-speed centrifugation (12k, 15 min, 4°C), the upper layer was taken as cytoplasmic fraction. Glassy, white nuclear pellets were resuspended in SDS-containing sample buffer and then boiled for western blot experiments.

## Yeast RNA and cDNA preparation

Yeast RNAs were prepared as previously described (*Oh et al., 2018*). Five to 10 OD yeast cells were taken from YPgly cultures and were washed with 0.1% DEPC treated water. Washed pellets were resuspended in 400 µL of AE buffer (50 mM sodium acetate pH5.3 and 10 mM EDTA). Forty microliters of 10% SDS was added to AE buffer resuspended cells and vortexed. Four hundred and forty microliters of phenol pH 8.0 (Sigma) was added to tubes, and then tubes were incubated at 65°C for 4 min. Tubes were cooled down in pre-chilled ice block for 2 min and were then centrifuged at 11,000 rpm for 2 min. Aqueous phase was transferred into new tubes. RNAs in the aqueous phase were prepared using phenol/chloroform extraction followed by ethanol precipitation. Potential genomic DNA contamination was removed by using DNA-free kit (ThermoFisher). One microgram of purified RNA and 0.5 µg of oligo dT primers were used for cDNA construction using ImProm II reverse transcription system (Promega) following manufacturer's instruction.

## In vitro kinase assays

All in vitro kinase assays were done in NEBuffer for protein kinase (NEB) at 30°C. As a phosphate donor, 5 mM ATP or GTP was used. Recombinant Snf1 catalytic domain (Snf1-CAT) was phosphorylated by 1.4 pmol of recombinant human CaMKK2 (Abnova) for 2 hr at 30°C for its activation (*Figure 5F*). When Tda1 fragments were used as kinases (*Figure 4F*), additional DTT was supplemented up to 1 mM. The reactions were quenched by SDS sample buffer addition, then boiled at 100°C for 5 min.

## MudPIT analysis

Trichloroacetic acid (TCA) precipitated proteins were urea-denatured, reduced, alkylated, and digested with endoproteinase Lys-C (Promega) followed by modified trypsin (Promega) as previously described (*Florens and Washburn, 2006*; *Washburn et al., 2001*). Peptide mixtures were loaded onto 100 µm fused silica microcapillary columns packed with 5 µm $C_{18}$ reverse phase (Aqua, Phenomenex), strong cation exchange particles (Partisphere SCX, Whatman), and reverse phase (*MacCoss et al., 2002*). Loaded microcapillary columns were placed in-line with a Quaternary Agilent 1260 series HPLC pump and a Velos Orbitrap mass spectrometer equipped with a nano-LC electrospray ionization source (ThermoFinnigan). Fully automated 10-step MudPIT runs were carried out on the electrosprayed peptides, as previously described (*Florens and Washburn, 2006*). Tandem mass spectra were interpreted using ProLuCID (*Eng et al., 1994*) against a database of 8956 sequences, consisting of 4303 *E. coli* proteins (downloaded from NCBI on 2013-11-06), 177 usual contaminants such as human keratins, IgGs, and proteolytic enzymes, 4476 'shuffled' sequences, and four yeast proteins sequences (Tda1, Snf1, Cka1, and a recombinant Tda1 C-terminal sequence from aa residues 354–586). Peptide/spectrum matches were sorted and selected using DTASelect (*Tabb et al., 2002*) with the following criteria set: spectra/peptide matches were only retained if they had a DeltCn of at least 0.08 and minimum XCorr of 1.8, 2.0, and 3.0 for singly, doubly, and triply charged spectra, respectively. In addition, peptides had to be fully tryptic and at least six amino acids long. Combining all runs, proteins had to be detected by at least two such peptides, or 1 peptide with two independent spectra. Under these criteria, the final FDRs at all levels were zero. Peptide hits from multiple runs were compared using CONTRAST (*Tabb et al., 2002*). To estimate relative protein levels, distributed normalized spectral abundance factors were calculated for each detected protein, as previously described (*Florens et al., 2006*; *Paoletti et al., 2006*; *Zybailov et al., 2006*).

## Acknowledgements

We thank Dr. Michael Church for scientific editing of the manuscript. We thank the Workman Lab members and Stowers core facilities for support during this project. These studies were supported by funds from the Stowers Institute and the National Institutes of General Medical Sciences grant R35GM118068 (JLW). Original data underlying this manuscript can be accessed from the Stowers Original Data Repository at http://www.stowers.org/research/publications/libpb-1536.

## Additional information

### Competing interests

Jerry L Workman: Reviewing editor, *eLife*. The other authors declare that no competing interests exist.

### Funding

| Funder | Grant reference number | Author |
| --- | --- | --- |
| Stowers Institute for Medical Research | Workman Lab | Jerry L Workman |
| National Institute of General Medical Sciences | R35GM118068 | Jerry L Workman |

The funders had no role in study design, data collection and interpretation, or the decision to submit the work for publication.

### Author contributions

Seunghee Oh, Conceptualization, Formal analysis, Investigation, Methodology, Writing - original draft; Jaehyoun Lee, Investigation; Selene K Swanson, Validation, Methodology; Laurence Florens, Michael P Washburn, Supervision, Methodology; Jerry L Workman, Conceptualization, Formal analysis, Supervision, Funding acquisition, Writing - original draft, Writing - review and editing

### Author ORCIDs

Seunghee Oh (iD) https://orcid.org/0000-0002-6701-9473
Michael P Washburn (iD) http://orcid.org/0000-0001-7568-2585
Jerry L Workman (iD) https://orcid.org/0000-0001-8163-1952

### Decision letter and Author response

Decision letter https://doi.org/10.7554/eLife.64588.sa1
Author response https://doi.org/10.7554/eLife.64588.sa2

## Additional files

### Supplementary files

• Supplementary file 1. Tda1 C fragment phosphorylation by recombinant Snf1-CAT and Cka1. The MudPIT analysis results showing the Tda1 C fragment (Tda1C, Tda1 aa 354–586) phosphorylation sites from in vitro kinase assays using Tda1C only, Tda1C with Snf1-CAT (Tda1C + rSnf1-CAT), and Tda1C with Cka1 (Tda1C + rCka1), respectively. Snf1-CAT was activated by pre-incubation with human CaMKK2 before the kinase assay with Tda1C. The Tda1 phosphorylation sites by Snf1-CAT are classified into three groups (I, II, and III) by the proximity of phosphorylation sites. (Total: total peptides detected, Modified: the number of phosphorylation containing peptides, PTM%: the percentage of modified peptide compared to total peptide detected.)

• Supplementary file 2. Yeast strains used in this study.

• Transparent reporting form

### Data availability

Original data underlying this manuscript can be accessed from the Stowers Original Data Repository at http://www.stowers.org/research/publications/libpb-1536.

The following dataset was generated:

| Author(s) | Year | Dataset title | Dataset URL | Database and Identifier |
| --- | --- | --- | --- | --- |
| Seunghee Oh | 2020 | LIBPB-1536 | http://www.stowers.org/ | Stowers Original Data |

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
