## [Decision Letter]

**Acceptance summary:**

This paper does an excellent job demonstrating that AMPK and CK2 signaling converge on histone kinase Tda1 to link external low glucose stress to chromatin regulation. The insight provided into how Tda1 is a novel mediator of H3T11P is exciting. This work is of wide interest to the chromatin, transcription, nutrient signaling community.

**Decision letter after peer review:**

Thank you for submitting your article "Yeast Nuak1 phosphorylates histone H3 threonine 11 in low glucose stress by the cooperation of AMPK and CK2 signaling" for consideration by *eLife*. Your article has been reviewed by three peer reviewers, including Xiaobing Shi as the Reviewing Editor and Reviewer #1, and the evaluation has been overseen by Jessica Tyler as the Senior Editor. The following individual involved in review of your submission has agreed to reveal their identity: Brian D Strahl (Reviewer #3).

The reviewers have discussed the reviews with one another and the Reviewing Editor has drafted this decision to help you prepare a revised submission.

Summary:

This is an excellent study from the Workman group that delineates the mechanism by which H3T11 becomes phosphorylated during nutrient (i.e., glucose) deprivation in yeast to regulate gene transcription. Previous studies from the Workman lab and others have revealed that H3T11 is phosphorylated from yeast to humans, and that this modification is important for transcriptional regulation. Specifically in yeast, the Workman group recently demonstrated that pyruvate kinase and CKII are required for H3T11P; however, whether additional enzymes were also involved and the interplay between kinases in mediating this mark was not well understood. As CKII was determined to not be the main kinase for H3T11P under glucose starvation, the Workman group sought to identify the relevant kinase(s). The authors found that Snf1 of the AMPK signaling pathway is required for proper H3T11P, but that not all of the regulatory subunits of AMPK (β and γ) are needed, thereby suggesting a novel connection of Snf1 with the terminal kinase. A search for this kinase found, surprisingly, that Tda1 (yeast NUAK1 homolog) is the bona fide kinase that mediates this modification. They show that Tda1 is activated by the nutrient regulated kinase Snf1, they identify the sites phosphorylated by Snf1, and they show that the nuclear translocation of Tda1 is dependent on Cka1 signaling. In all, this work describes an elegant molecular pathway wherein multiple singling pathways and kinases converge to regulate a key kinase that mediates an important chromatin modification required for glucose-deprived gene regulation.

Overall, this study is thorough, and the conclusions are strong. This paper does an excellent job demonstrating which kinases are responsible for phosphorylating which sites. The insight provided into how Tda1 is a novel mediator of H3T11P is exciting, as well as the interesting connections made to its nuclear localization. Taken together, this work would be of wide interest to the chromatin, transcription, nutrient signaling community. Therefore, I support its publication in *eLife* with minor modifications.

Essential revisions:

1) A major weakness in this story is the lack of any indication that phosphorylation of histone H3T11 is biologically significant. Do cells lacking this signaling pathway or tda1 deletion have any phenotypes or defect in gene expression? Some experimental data to demonstrate that phosphorylation of H3T11 is important for some aspects of nutrient stress response would increase the significance of this paper tremendously.

2) It is interesting that Snf1-mediated phosphorylation of S483/T484 is critical for Tda1 kinase activity. However, the in vivo function of this phosphorylation event is not very clear. It will help to complete the circuit in the paper if it can be experimentally tested if mutation of S483/T484 leads to defective gene activation of glucose sensitive target genes. It would be assumed but not formally shown. Also, it would also be interesting to see if a phospho-memetic (S/T to E) mutation of Tda1 might bypass the need for Snf1/CKII, etc.

3) Similar experiments to test the role of CK2-mediated Tda1 phosphorylation would also strength the paper greatly.

4) The conclusion that Snf1 is a kinase that can phosphorylate H3S10 is arguable. The authors may want to provide more biochemical data using recombinant proteins and orthogonal approaches, or tune down their conclusion.

---

## [Author Response]

Essential revisions:1) A major weakness in this story is the lack of any indication that phosphorylation of histone H3T11 is biologically significant. Do cells lacking this signaling pathway or tda1 deletion have any phenotypes or defect in gene expression? Some experimental data to demonstrate that phosphorylation of H3T11 is important for some aspects of nutrient stress response would increase the significance of this paper tremendously.

We thank the reviewer for this comment. In our previous work, also published in e*Life* (PMID: 29938647), we reported genome-wide ChIP-seq results of H3pT11 in low glucose YPgly media (GSE111218). We also reported RNA-Seq results of several mutants whose H3pT11 was impaired (i.e. H3T11A mutant) in YPgly media (GSE111217). In that manuscript, we reported a solid relationship between the transcription activation of low glucose stress responsive genes and H3pT11 signal increase upon media shift from YPD to YPgly media. In addition, we found the significant transcriptional defects of several low glucose stress responsive genes in H3T11A mutant. In this regard, we believe that we successfully proved a biological significance of H3pT11 in low glucose condition in our previous work. Thanks to the reviewer’s comment we realize that we need to better review the biological significance demonstrated in our previous study in the Introduction of this manuscript. We have added a summary of the previous manuscript to the revised manuscript (Introduction).

In addition, as we already know the target genes of H3pT11, we selected several genes whose transcription are impaired in H3T11A mutant in low glucose stress. We tested their gene expression levels in tda1Δ mutants to see if tda1Δ showed a similar phenotype compared to H3T11A mutant. We added a new Figure 6—figure supplement 2, describing this result.

2) It is interesting that Snf1-mediated phosphorylation of S483/T484 is critical for Tda1 kinase activity. However, the in vivo function of this phosphorylation event is not very clear. It will help to complete the circuit in the paper if it can be experimentally tested if mutation of S483/T484 leads to defective gene activation of glucose sensitive target genes. It would be assumed but not formally shown. Also, it would also be interesting to see if a phospho-memetic (S/T to E) mutation of Tda1 might bypass the need for Snf1/CKII, etc.

In newly made Figure 6—figure supplement 2, we compared the expression levels of four H3pT11 target genes (HSP26, GRE1, GND2, and SOL4) in WT, tda1Δ, and Tda1 S483A/T484A (AA) mutant. The data clearly showed that AA mutant failed to restore H3pT11 target gene expression levels compared to WT. Indeed, AA mutant showed a similar level of transcription to tda1Δ mutant. The AA mutant did not show transcription defects at non-target genes (PMA1 and MET13). These results suggest the critical role of Tda1 S483/T484 phosphorylation in the regulation of Tda1 activity.

Based on the reviewer’s suggestion, we made Tda1 S483D/T484E (DE) mutant and tested whether the potential phosphorylation mimic mutant could bypass the requirement of Snf1 or CK2. Interestingly, we found that DE mutant, as well as AA mutant could not restore H3pT11. The result is shown in Author response image 1.

**Author response image 1. sa2fig1:** H3pT11 levels of Tda1 WT, Tda1 S483A/T484A, and Tda1 S483D/T484E mutant expressing strains analyzed by western blot. Cells were taken from overnight saturated media.

Although we showed that Tda1 phosphorylation at S483/T484 is critical for the regulation of Tda1 activity (Figure 6D), we do not know the exact mechanism how the modification controls Tda1 activity. This needs further investigation in the near future. If we can understand the detailed mechanism of Tda1 activity control by phosphorylation, then we may understand why the DE mutant failed to mimic Tda1 phosphorylation.

3) Similar experiments to test the role of CK2-mediated Tda1 phosphorylation would also strength the paper greatly.

We appreciate the reviewer’s attention on this point. We showed that the Tda1 C terminal was phosphorylated by Cka1 at serine 578 (Figure 6A and Supplementary file 1) using in vitro kinase assays. We tested Tda1 S578A mutant for H3pT11 and found that the Tda1 S578A mutant did not show any defect in H3pT11 (Figure 6B), suggesting that Cka1 may not regulate Tda1 by direct phosphorylation at serine 578. Thus far, we were not able to find any evidence that Cka1 could phosphorylate Tda1 at sites other than S578, or control Tda1 nuclear localization by direct phosphorylation. The mechanism how CK2 regulates Tda1 nuclear localization needs more investigation in the near future.

4) The conclusion that Snf1 is a kinase that can phosphorylate H3S10 is arguable. The authors may want to provide more biochemical data using recombinant proteins and orthogonal approaches, or tune down their conclusion.

We agree with reviewers comment that Snf1 as a H3S10 kinase is arguable. As H3pS10 is dispensable for H3pT11 onset upon low glucose stress (Figure 1E and F), we believe that Snf1 as a histone H3 S10 kinase is not a critical part of this study. In this regard, we revised manuscript as below.

From “Snf1 phosphorylates H3 at serine 10 at the INO1 gene promoter…”

to “Snf1 is required for H3 serine 10 phosphorylation (H3pS10) at the INO1 gene promoter…”

From “Snf1 can phosphorylate histone H3 at serine 10 in yeast.”

to “Snf1 is required for H3pS10 at the INO1 gene promoter in yeast.”

In addition, we changed Figure 4D. in the previous Figure 4D, we showed that yeast flag purified Snf1 could phosphorylate both H3S10 and H3T11 in vitro. To avoid confusion, we repeated the experiment without flag purified Snf1. Instead, we added GST-Cka1 as a new control.